# Differences in the genome, methylome, and transcriptome do not differentiate isolates of *Streptococcus equi* subsp. *equi* from horses with acute clinical signs from isolates of inapparent carriers

Ellen Ruth A. Morris[1], Ashley G. Boyle[2], Miia Riihimäki[3], Anna Aspán[3], Eman Anis[4], Andrew E. Hillhouse[5,6], Ivan Ivanov[7], Angela I. Bordin[1], John Pringle[3], Noah D. Cohen[1]*

1 Department of Large Animal Clinical Sciences, College of Veterinary Medicine & Biomedical Sciences, Texas A&M University, College Station, Texas, United States of America, 2 Department of Clinical Studies, School of Veterinary Medicine, University of Pennsylvania, New Bolton Center, Kennett Square, Pennsylvania, United States of America, 3 Department of Clinical Sciences, Swedish University of Agricultural Sciences, Uppsala, Sweden, 4 Department of Pathobiology, School of Veterinary Medicine, University of Pennsylvania, New Bolton Center, Kennett Square, Pennsylvania, United States of America, 5 Department of Veterinary Pathobiology, College of Veterinary Medicine & Biomedical Sciences, Texas A&M University, College Station, Texas, United States of America, 6 Texas A&M Institute for Genome Sciences and Society, College of Veterinary Medicine & Biomedical Sciences, Texas A&M University, College Station, Texas, United States of America, 7 Department of Veterinary Physiology and Pharmacology, College of Veterinary Medicine & Biomedical Sciences, Texas A&M University, College Station, Texas, United States of America

* ncohen@cvm.tamu.edu

**Data Availability Statement:** All whole genome sequencing data were uploaded to NCBI's GenBank

## Abstract

*Streptococcus equi* subsp. *equi* (SEE) is a host-restricted bacterium that causes the common infectious upper respiratory disease known as strangles in horses. Perpetuation of SEE infection appears attributable to inapparent carrier horses because it neither persists long-term in the environment nor infects other host mammals or vectors, and infection results in short-lived immunity. Whether pathogen factors enable SEE to remain in horses without causing clinical signs remains poorly understood. Thus, our objective was to use next-generation sequencing technologies to characterize the genome, methylome, and transcriptome of isolates of SEE from horses with acute clinical strangles and inapparent carrier horses—including isolates recovered from individual horses sampled repeatedly—to assess pathogen-associated changes that might reflect specific adaptions of SEE to the host that contribute to inapparent carriage. The accessory genome elements and methylome of SEE isolates from Sweden and Pennsylvania revealed no significant or consistent differences between acute clinical and inapparent carrier isolates of SEE. RNA sequencing of SEE isolates from Pennsylvania demonstrated no genes that were differentially expressed between acute clinical and inapparent carrier isolates of SEE. The absence of specific, consistent changes in the accessory genomes, methylomes, and transcriptomes of acute clinical and inapparent carrier isolates of SEE indicates that adaptations of SEE to the

and Sequence Read Archive (SRA) under BioProject PRJNA704656 (https://www.ncbi.nlm.nih.gov/bioproject/?term=PRJNA704656). The RNA sequencing data were uploaded to NCBI's Gene Expression Omnibus (GEO) which are available by the accession number GSE167862 (https://www.ncbi.nlm.nih.gov/geo/query/acc.cgi?acc=GSE167862). For the specific accession number for each individual genome, please see Supplemental S1 to S7 Tables.

**Funding:** Funding for the whole genome sequencing of the isolates was provided by the Link Equine Research Endowment, College of Veterinary Medicine & Biomedical Sciences, Texas A&M University. Funding for the RNA-Seq portion of the study was provided by a grant from the Department of Large Animal Clinical Sciences, College of Veterinary Medicine & Biomedical Science, Texas A&M University. Dr. Cohen is supported by the Patsy Link Chair in Equine Research. Funding for the Swedish contribution to this work was provided by the Swedish Research Council for Environment, Agricultural Sciences and Spatial Planning (FORMAS) grant # 221-2013-606. The funders had no role in study design, data collection and analysis, decision to publish, or preparation of the manuscript.

**Competing interests:** No authors have competing interests.

host are unlikely to explain the carrier state of SEE. Efforts to understand the carrier state of SEE should instead focus on host factors.

## Introduction

*Streptococcus equi* subsp. *equi* (SEE) is a host-specific bacterial pathogen that causes the infectious disease of horses known as strangles [1–5]. Infection with SEE occurs primarily in the upper respiratory tract, and is very contagious with a high rate of morbidity in naïve horses [3]. Typically, infection results in lethargy, pyrexia, swollen lymph nodes, guttural pouch empyema, and nasal discharge [3, 4]. Other clinical signs of disease can be observed, including dissemination of infection to other organs and immune-mediated sequelae such as vasculitis and myositis [3, 5]. Strangles is an ancient disease that is prevalent among horses worldwide [1, 6]. The persistence of the disease appears to be attributable to the ability of SEE to survive in horses that are infected but do not show clinical signs. SEE cannot survive in the external environment for extended periods of time: SEE can persist approximately 2 days on surfaces outside its host [7], and from 1 to 4 weeks in a wet environment, dependent upon the season [8]. There are no known biological or mechanical vectors of SEE [3], and horses that have recovered from the disease usually develop prolonged immunity [3, 4]. Consequently, the most likely source of spread and persistence of SEE is horses that appear healthy but shed SEE undetected (so-called **inapparent carrier horses**) [3, 9, 10]; these carriers transmit SEE to susceptible horses, thereby perpetuating the disease in nature [11, 12].

Several host- and pathogen-associated adaptations have been suggested to give rise to the capacity for SEE to evade the immune system and persist within the host. The ability for some strains to be carried by apparently healthy horses has been attributed to the presence of chondroids (*i.e.*, concretions of inspissated pus) or empyema in the guttural pouches of infected horses recovered from strangles [9, 11]. However, cases have been documented in which no clinical signs or vestiges of inflammation or niduses of infection (such as chondroids) were noted from clinically inapparent carriers of SEE [13, 14]. Truncation of the N-terminus of the M-like protein (SeM) has been hypothesized to contribute to the ability of SEE to remain undetected in the host [15]. Another factor that has been proposed to contribute to inapparent carriage of SEE in horses is its equibactin locus (*eqbA*–*eqbN*), the novel iron acquisition element present on the integrative and conjugative element (ICE), ICE*Se2* [10, 16]. More efficient iron acquisition is theorized to aid in the ability of SEE to better survive in the host without inducing clinical signs [16]. Despite these proposed characteristics of carrier isolates, none has been documented to be identified consistently among isolates of SEE from inapparent carriers. Consequently, it is unclear whether the inapparent carrier state of SEE is attributable to agent factors (adaptations to the host), host factors (such as immunity), or both.

Next-generation sequencing (NGS) technologies such as whole genome sequencing (WGS) or RNA sequencing (RNA-Seq) of SEE can be employed to investigate agent-associated adaptions within the bacterial genome or transcriptome that contribute to inapparent carriage. Using WGS, the bacterial genome can be defined by elements that make up the core or accessory genome [17, 18]. Core genome elements (CGE) are those found in the genomes of most isolates of the same bacterial species, whereas accessory genome elements (AGE) are elements that are not found in all isolates of the same bacterial species. Comparison of the AGE has

been used to identify differences among isolates from the same bacterial species collected from different environments [17]. Additionally, using PacBio single molecule, real-time (SMRT) WGS allows for characterization of the methylation patterns of bacterial genomes [19]. Methylation of their DNA protects bacteria against bacteriophage or other foreign DNA; methyl groups sharing the same sequence motif as the bacteria's own DNA protect against enzymatic degradation, whereas the DNA lacking the same methylation is recognized as foreign by endonucleases that cleave at these unmethylated motifs [20, 21]. Methylation can also alter gene expression and even alter virulence in some bacteria [22–24]. Methylated bacterial DNA is most commonly recognized as residues of N6-methyl-adenosine (m6A), N4-methyl-cytosine (m4C), or C5-methyl-cytosine (m5C) [20, 21]. In addition to the genome and the methylome, assessing the transcriptome through RNA-Seq can be used to characterize changes in gene expression that influence phenotype of the organism. For example, RNA-Seq revealed that differing regulation of gene expression resulted in a change in SEE colony morphology [25]. Thus, RNA-Seq might distinguish strains of SEE that result in inapparent carriage from isolates obtained from horses with acute clinical signs.

To our knowledge, however, potential differences in the genome, methylome, and transcriptome of inapparent carrier and acute clinical strains of SEE has not been investigated. Thus, we aimed to compare the AGE, methylomes, and transcriptomes of strains of SEE recovered from horses from within the same geographical regions that recovered from SEE without clinical signs (inapparent carriers) with strains of SEE from those with acute clinical signs of strangles, including some isolates collected by sequential sampling of individual horses. The purpose of these comparisons was to identify evidence of any adaptions of the pathogen to its host. We showed that there were no consistent differences between the 2 phenotypes of SEE strains for the AGE, methylome, or transcriptome that might explain persistence in the host. These findings indicate that pathogen-associated adaptions are highly improbable as an explanation for the ability of SEE to go undetected and persist within its host.

## Material and methods

### *Streptococcus equi* subsp. *equi* isolates

Carrier and clinical SEE isolates from Pennsylvania (PA-USA) were provided by a co-author (AGB), and sequence data of Swedish isolates of SEE predominantly from acute clinical cases and their isolates after progression to inapparent carriers were provided by 2 other co-authors (MR and JP) (Table 1). For the purposes of our study, inapparent carriers were defined as horses either recovered from strangles or exposed to strangles cases that were absent of clinical signs for $\geq$ 6 weeks prior to collection of the isolate. Swedish isolates of SEE (n = 14) were from a single outbreak at an individual farm in Sweden previously described [13] comprised of 8 isolates from inapparent carriers and 6 isolates from those with clinical disease; 5 horses from this herd contributed isolates during both acute disease and the inapparent carrier state. No abnormalities were noted during endoscopy of the guttural pouches of the Swedish horses that were inapparent carriers with the exceptions of 1 horse with chondroids (489_010) and another horse in which a moderate amount of mucus (but no purulent exudate) was noted in the right guttural pouch only (489_007) (S1 Table). Isolates of SEE from PA-USA (n = 21) were from 11 inapparent carriers and 10 acute clinical cases located in a similar geographical area of the state, and isolates spanned different years (2014 to 2017). Results of guttural pouch endoscopy were available for only 5 of 11 (45%) horses from PA-USA from which carrier strains were recovered; all 5 horses had abnormal findings within their guttural pouches (S1 Table).

**Table 1. Description of the 14 SEE isolates from Sweden and the 21 SEE isolates from Pennsylvania.**

| Genome ID | Location | Status | Horse ID | Collection Source | Collection Date | Duration From Resolution of Clinical Signs | ST | SeM |
|---|---|---|---|---|---|---|---|---|
| 470_007 | Sweden | Carrier | H1 | NL | 11/11/2015 | 20 weeks | 179 | 72 |
| 470_006 | Sweden | Acute | H2 | NL | 5/21/2015 | NA | 179 | 72 |
| 470_003 | Sweden | Carrier | H2 | NL | 8/26/2015 | 12 weeks | 179 | 72 |
| 470_002 | Sweden | Acute | H3 | NL | 5/21/2015 | NA | 179 | 72 |
| 489_007 | Sweden | Carrier | H3 | NL | 11/11/2015 | 24 weeks | 179 | 72 |
| 470_001 | Sweden | Acute | H4 | NL | 5/21/2015 | NA | 179 | 72 |
| 489_004 | Sweden | Acute | H5 | NL | 6/6/2015 | NA | 179 | 72 |
| 470_008 | Sweden | Carrier | H5 | NL | 11/11/2015 | 20 weeks | 179 | 72 |
| 489_006 | Sweden | Carrier | H5 | NL | 11/11/2015 | 20 weeks | 179 | 150[a] |
| 489_003 | Sweden | Acute | H7 | NL | 5/21/2015 | NA | 179 | 72 |
| 489_010 | Sweden | Carrier | H7 | GPL | 3/3/2016 | 50 weeks | 179 | 152[a] |
| 489_002 | Sweden | Acute | H8 | NL | 5/21/2015 | NA | 179 | 72 |
| 489_005 | Sweden | Carrier | H8 | NL | 8/26/2015 | 12 weeks | 179 | 72 |
| 489_009 | Sweden | Carrier | H8 | GPL | 3/3/2016 | 50 weeks | 179 | 151 |
| 20–080 | Pennsylvania | Carrier | PA1 | GPL | 7/15/2014 | 6 weeks | 179 | 39 |
| 20–081 | Pennsylvania | Carrier | PA2 | GPL | 8/20/2014 | 12 weeks | 179 | 39 |
| 20–082 | Pennsylvania | Carrier | PA3 | GPL | 11/26/2014 | 20 weeks | 179 | 39 |
| 20–083 | Pennsylvania | Carrier | PA4 | GPL | 12/3/2014 | 20 weeks | 179 | 39 |
| 20–084 | Pennsylvania | Carrier | PA5 | NL | 7/27/2016 | 16 weeks | 179 | 28 |
| 20–085 | Pennsylvania | Carrier | PA6 | NL | 12/5/2016 | None | 179 | 147 |
| 20–086 | Pennsylvania | Carrier | PA7 | NL | 7/27/2016 | 8 weeks | 179 | 39 |
| 20–087 | Pennsylvania | Carrier | PA8 | NL | 1/11/2017 | 8 weeks | 179 | 224 |
| 20–088 | Pennsylvania | Carrier | PA9 | GPL | 4/4/2017 | 12 weeks | 179 | 147 |
| 20–089 | Pennsylvania | Carrier | PA10 | GPL | 5/17/2017 | None | 179 | 225 |
| 20–090 | Pennsylvania | Carrier | PA11 | GPL | 8/8/2017 | 7 weeks | 179 | 226 |
| 20–091 | Pennsylvania | Acute | PA12 | GPL | 6/6/2014 | NA | 179 | 28 |
| 20–092 | Pennsylvania | Acute | PA13 | GPL | 4/24/2014 | NA | 179 | 227 |
| 20–093 | Pennsylvania | Acute | PA14 | NL | 2/16/2017 | NA | 179 | 224 |
| 20–094 | Pennsylvania | Acute | PA15 | NL | 8/27/2014 | NA | 179 | 28 |
| 20–095 | Pennsylvania | Acute | PA16 | NL | 2/1/2016 | NA | 179 | 39 |
| 20–096 | Pennsylvania | Acute | PA17 | NL | 3/10/2014 | NA | 179 | 228 |
| 20–097 | Pennsylvania | Acute | PA18 | NL | 3/17/2014 | NA | 179 | 28 |
| 20–098 | Pennsylvania | Acute | PA19 | NL | 2/17/2016 | NA | 179 | 28 |
| 20–099 | Pennsylvania | Acute | PA20 | NL | 3/4/2016 | NA | 179 | 28 |
| 20–100 | Pennsylvania | Acute | PA21 | NL | 3/24/2016 | NA | 179 | 28 |

ST, Sequence type; SeM, M-like protein; NL, Nasopharyngeal lavage; GPL, Guttural pouch lavage; NA, Not applicable.

[a]Truncation noted in SeM protein.

## Bacterial DNA extraction and whole genome sequencing

The PA-USA SEE isolates were cultured overnight in 3 ml of Todd Hewitt broth (THB; HIME-DIA®, West Chester, PA, USA) in 5% $CO_2$ at 37°C. Following incubation overnight, bacterial isolates were centrifuged at 3,000 x g for 10 minutes to create a pellet. The supernatants were discarded, and DNA extractions were performed using the DNeasy® UltraClean® Microbial kit (Qiagen®, Hilden, Germany), following the manufacturer's instructions with some modifications. Briefly, the bacterial pellets were resuspended in 300 μl of PowerBead solution, and transferred into PowerBead tubes. Fifty μl of solution SL was added, and the PowerBead tubes

were incubated at 70˚C for 10 minutes, followed by horizontal vortexing for an additional 10 minutes. Then, the PowerBead tubes were centrifuged and the supernatants were transferred to new tubes. One hundred µl of solution IRS was added to the supernatants, incubated for 15 minutes at 4˚C, and then centrifuged. The supernatants were transferred to another tube without disturbing the pellet, and 900 µl of solution SB was added. Seven hundred µl of this solution was transferred to MB spin column tubes, centrifuged, and, after the flow-through was discarded, this step was repeated. Additionally, 300 µl of solution CB was added to the columns and centrifuged. Another centrifuge step was performed to remove any excess fluid, and the MB spin columns were transferred to new collection tubes. Finally, 50 µl of the solution EB was added to the columns and centrifuged. The quality and concentration of the DNAs were assessed using a NanoDrop spectrophotometer (ND-1000, Thermo Fisher Scientific, Waltham, MA, USA), and sent to the Duke Center for Genomic and Computational Biology (GCB) for WGS using the PacBio Sequel platform.

The Swedish SEE isolates, cultured from horses during a strangles outbreak as described by Riihimäki et al., [13] were retrieved from storage at -70˚C, subculture was performed, and then grown overnight on 15-cm-diameter blood agar plates (SVA, Uppsala, Sweden) in 5% $CO_2$ at 37˚C. DNAs were extracted by the Genomic-tip 100/G kit (GT) (Qiagen, Hilden, Germany) according to the manufacturer's protocol, but bacterial lysis was performed prior to extraction to obtain high molecular weight DNA. Briefly, SEE growth from the agar plates were harvested by a 10-µl loop into a 2-ml tube and thereafter lysed in 200 µl of 50 mM EDTA pH 8.0 supplemented with 20 µl (100 mg/ml) lysozyme. After incubation on a thermomixer for 4 hours at 37˚C / 400 x g, 400 µl GT buffer B1 (provided by the manufacturer of the kit) and 20 µl proteinase K were added, and samples were mixed by inverting the tubes 10 times. This was followed by a further incubation for 4 hours, at 54˚C / 400 x g. Samples were frozen at -80˚C overnight, flash-thawed at 50˚C, and 300 µl of GT buffer B2 was added. Again, samples were mixed by inverting the tubes 10 times. Five µl of RNase was added and after 10 minutes at room temperature, samples were mixed for 30 minutes at 50˚C / 400 x g, before DNA extraction. After DNA extraction, the DNA quality was assessed using a NanoDrop spectrophotometer (ND-8000, Thermo Fisher Scientific, Waltham, MA, USA), and concentrations were determined using a Qubit® 2.0 fluorometer (Invitrogen, Carlsbad, CA, USA). The DNA from the 14 Swedish SEE isolates were then sent to the SciLifeLab (https://www.scilifelab.se/) for PacBio sequencing.

## Bacterial RNA extraction and RNA sequencing

Carrier and clinical PA-USA SEE isolates were grown in THB for 4 hours (exponential phase growth) at 37˚C in 5% $CO_2$. Following the 4-hour incubation, liquid cultures were centrifuged at 3,000 x g for 10 minutes to pellet the bacterium and the supernatants were discarded. The bacterial RNAs were then extracted using the RiboPure™ RNA Purification kit (Ambion® RiboPure™-Bacteria Kit; Invitrogen™, Carlsbad, CA, USA) following the manufacturer's instructions. Briefly, the SEE pellets were resuspended in 350 µl of the $RNA_{WIZ}$ solution, and then transferred to tubes with Zirconia beads. The tubes were placed on a horizontal vortex adaptor, beat for 10 minutes at maximum speed, and then centrifuged at 13,000 x g for 5 minutes at 4˚C. The supernatants containing the lysed bacteria were transferred to fresh tubes, 0.2 volumes of chloroform were added, and samples were incubated for 10 minutes at room temperature. To separate the organic and aqueous phases, tubes were centrifuged for 5 minutes at 4˚C. The aqueous phases were transferred to new tubes, 0.5 volumes of 100% ethanol were added, mixed thoroughly, and transferred to filter cartridges in 2-ml tubes. The filter cartridge tubes were then centrifuged for 1 minute, the flow-through discarded, and the filters were

washed by the addition of 700 μl of Wash Solution 1. A second and third wash steps were performed with the addition of Wash Solution 2/3. After the third wash step, the filter cartridges were transferred to new tubes. Finally, the RNA was eluted by 50 μl of Elution Solution, and a DNase treatment was performed. The quality and purity of the RNAs were assessed using the NanoDrop (ND-1000, Thermo Fisher Scientific, Waltham, MA, USA).

At the Texas A&M Institute for Genome Sciences and Society (TIGSS) molecular genomics laboratory, RNA extracted from the 21 PA-USA SEE isolates were quantified using the Qubit fluorometric RNA (Thermo Fisher Scientific, Waltham, MA, USA) assay for normalization prior to library preparation. RNA libraries were prepared using the Stranded Total RNA Preparation kit (Illumina©, San Diego, CA, USA) following the manufacturer's instructions, in which each isolate received a unique barcode. The 21 isolates were pooled, and RNA-Seq was performed on the NovaSeq 6000 (Illumina©, San Diego, CA, USA) instrument that generated 150-base-pair, paired-end sequences. The sequencing run produced approximately 6 million reads per sample and resulted in ~200 X coverage for each sample.

## Bioinformatic analysis

Following WGS of the PA-USA and Swedish isolates, the Texas A&M High Performance Research Computing (HPRC) clusters were used to assemble genomes *de novo* using CANU (v1.7) [26], with the parameters of increased coverage (*corOutCoverage* = 100) and increased assembly sensitivity (*corMhapSensitivity* = high). Assembled genomes were confirmed to be SEE through the ribosomal multilocus sequence types database [27], and StrainSeeker [28]. The ST- and SeM-type of each of the assembled genomes of SEE were determined using the PubMLST *Streptococcus zooepidemicus* database [29, 30]. Then, assembled genomes were annotated using RASTtk (v2.0) [31] via the web-based server. Following annotation, the annotated genomes were inputted into Spine (v0.3.2) [17] to define the core genome (*i.e.*, elements found in all genomes) of SEE. Using the core genome output from Spine, the AGE (*i.e.*, elements found present in some genomes but absent from others) were identified using AGEnt (v0.3.1) [17]. Finally, ClustAGE (v0.8) [32] was implemented to identify and group the AGE that differ within the carrier and clinical SEE isolates. A graphical representation of clustered AGE for each individual genome was generated with the ClustAGE plot (http://vfsmspineagent.fsm.northwestern.edu/cgi-bin/clustage_plot.cgi). AGE were only included if ≥ 95% of the protein was identified. Comparisons of the AGE of carrier and clinical SEE were performed using custom R scripts (S1 Appendix). We conducted separate AGE analyses for SEE isolates from Sweden and PA-USA to avoid potential confounding effects by geographical location. A phylogenetic tree was built to assess the relatedness of the Swedish SEE isolates using PATRIC (v3.6.9) with default parameters [33]. Multiple sequence alignment of the SeM nucleotide sequences was performed using Clustal Omega (v1.2.4) at EMBL-EBI [34, 35].

The complete methylation profiles of carrier and clinical SEE genomes were characterized with the BaseMod (https://github.com/ben-lerch/BaseMod-3.0) pipeline in the PacBio SMRT Link (v8.0) command line tools. Briefly, pbmm2 was used to align the raw sequence read BAM files to the reference genome (SEE 4047). Using the aligned BAM file outputs, the kineticTools function *ipdSummary* was implemented to generate a GFF and CSV files with base-modification information. Next, the MotifMaker *find* function was used to generate a second set of CSV files with identified consensus motifs. Finally, the execution of the MotifMaker *reprocess* function generated GFF files with all the modifications that were associated with motifs. Using R (v4.0.3), the motif GFF files were filtered based on the presence of a known methylation types (m4C or m6A), and a having QV score (a quality score for the detection event) of ≥ 30.

The filtered GFF files of carrier and clinical SEE genomes were annotated by the SEE 4047 reference genome with the BedTools (v2.29.2) [36] *annotate* function. The annotated outputs for both carrier and clinical SEE were then compared by looking for the presence or absence of methylation on proteins throughout the genomes using custom R scripts (S1 Appendix). Identified motifs were then compared to the SEE 4047 genome using the Restriction Enzyme Database (REBASE) [37].

Following sequencing of the RNA of PA-USA SEE isolates at TIGSS, using the HPRC clusters raw RNA reads had their quality checked using FastQC (v0.11.6; www.bioinformatics. babraham.ac.uk/projects/fastqc/), and low quality reads were trimmed using Trimmomatic (v0.36) [38]. These filtered reads were then aligned and quantified against the reference genome SEE 4047 using Salmon (v1.3.0) [39]. The transcriptomes of all carrier SEE isolates were compared to clinical SEE isolates with edgeR (v3.30.3) [40] to identify any significantly (false discovery rate [FDR] $\leq$ 0.05) differently expressed genes with a $\log_2$-fold change (logFC) of $\leq$ -1 or $\geq$ 1 using a quasi-likelihood negative binomial generalized log-linear model (S1 Appendix) [41].

## Accession numbers

Genomes and raw sequence files were submitted to NCBI's GenBank and Sequence Read Archive under BioProject PRJNA704656. The RNA-Seq transcripts were deposited to NCBI's Gene Expression Omnibus (GEO) and are accessible through the GEO Series accession number GSE167862 (https://www.ncbi.nlm.nih.gov/geo/query/acc.cgi?acc=GSE167862) [42]. The specific accession numbers for each genome can be found in S2 Table.

## Results

Initially, the WGS data were used to define the AGE of the SEE isolates. The AGE were examined to identify genetic elements that differed between SEE isolates collected from acute and inapparent carrier cases. No consistent or significant differences in AGE were observed between the carrier (n = 8) and acute clinical (n = 6) isolates of SEE from the Swedish outbreak (Fig 1). Similarly, there were no differences identified between AGE of the carrier (n = 11) and acute (n = 10) SEE isolates from PA-USA (Fig 2). Many components identified in the AGE were associated with acquired genetic elements. Markedly fewer AGE elements were identified in the Swedish SEE isolates (S3 Table) than in the PA-USA SEE isolates (S4 Table). The phylogenetic assessment of the Swedish SEE isolates demonstrated that there were minor genomic differences between isolates recovered from either clinical or carrier state from the same individual horses, but these adaptations were not consistent among individuals (S1 Fig). For example, 2 carrier isolates (489_006 [H5], 489_010 [H7]) from Sweden were noted to have a truncated SeM protein (Table 1). Although neither horse had this truncation identified during acute clinical infection, in 1 horse (H5) the truncated isolate was collected via nasopharyngeal lavage simultaneously with a non-truncated isolate. These truncations were found at the beginning of the SeM protein, but ended at nucleotide base 318 and 333 in isolate 489_006 and 489_010, respectively (S2 Appendix). Furthermore, no other truncation in the SeM proteins were identified in the remaining isolates of SEE from Sweden or PA-USA.

Because some methylation events have been described to influence gene expression in prokaryotes [22–24], we performed additional characterization of these bacterial genomes by examining the methylomes of the carrier and acute clinical strains of SEE. As done for the AGE sequence data, separate analyses of the global methylation patterns determined from PacBio WGS were performed for the Swedish and PA-USA isolates of SEE. In both Swedish and PA-USA SEE isolates, no differences in methylation patterns were observed that

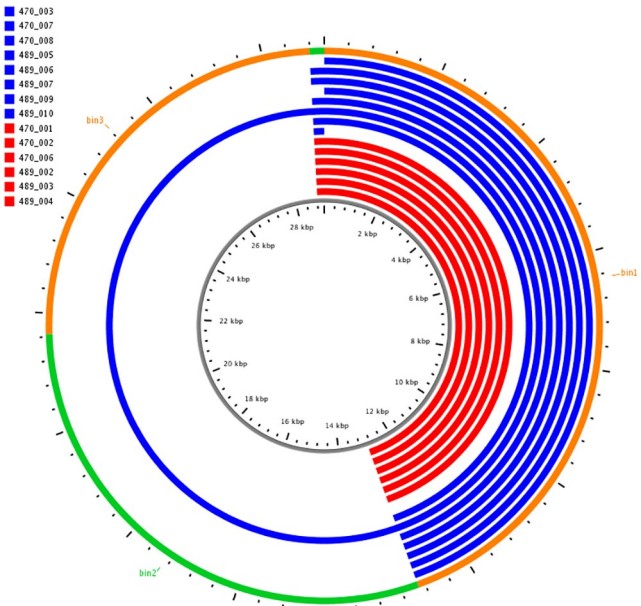

**Fig 1. Comparison of accessory genome elements (AGE) of inapparent carrier SEE (n = 8), and acute clinical SEE (n = 6) genomes from Sweden.** The outer ring shows the ClustAGE bins that are ≥ 200 base-pairs in size these are ordered clockwise from the largest bin to the smallest bin, and are differentiated by orange and green to define bin borders. The concentric inner bands show the distribution of AGE within each individual isolate. Bands that are blue represents inapparent carrier isolates, and bands that are red represent acute clinical isolates. The central ruler of the figure indicates the cumulative size of the AGE in kilobases.

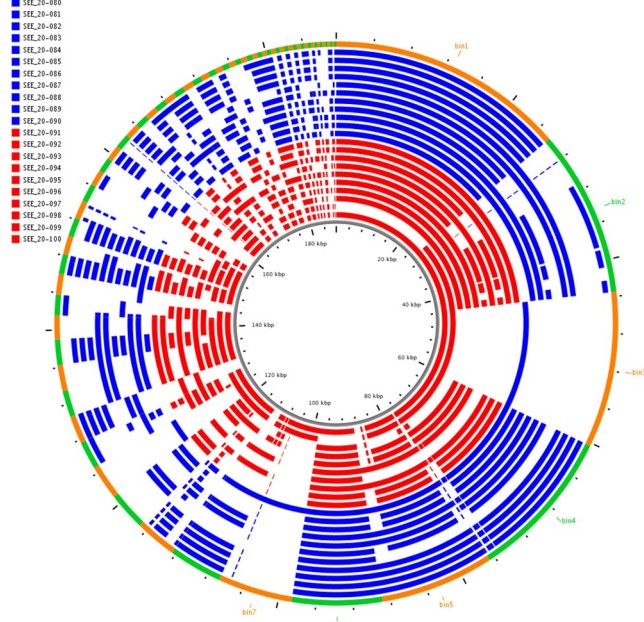

**Fig 2. Comparison of accessory genome elements (AGE) of inapparent carrier SEE (n = 11), and acute clinical SEE (n = 10) genomes from Pennsylvania.** The outer ring shows the ClustAGE bins that are ≥ 200 base-pairs in size these are ordered clockwise from the largest bin to the smallest bin, and are differentiated by orange and green to define bin borders. The concentric inner bands show the distribution of AGE within each individual isolate. Bands that are blue represents inapparent carrier isolates, and bands that are red represent acute clinical isolates. The central ruler of the figure indicates the cumulative size of the AGE in kilobases.

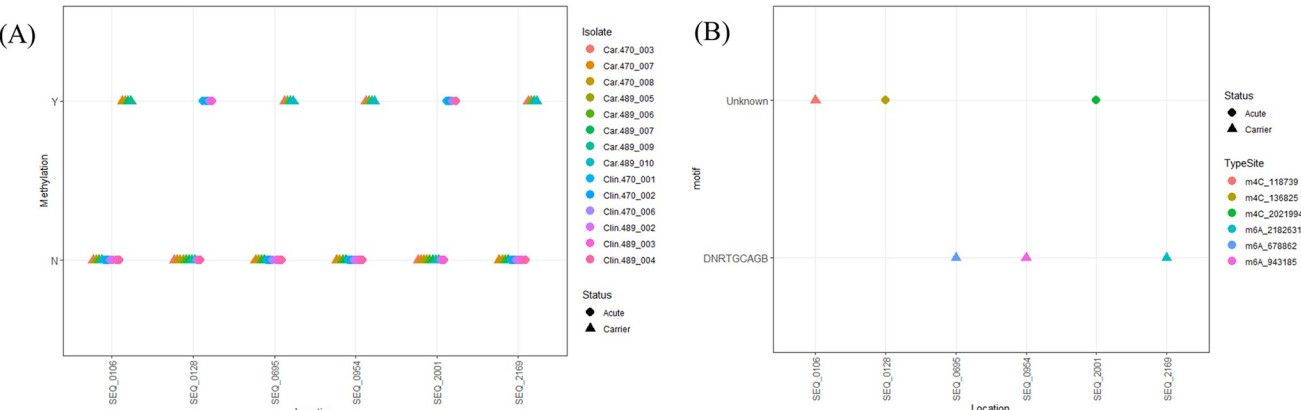

**Fig 3. Methylation locations and motifs from SEE isolates from Sweden.** (A) Depiction of whether methylation occurred at a specified genomic location. Genomic locations are indicated along the x-axis, and whether methylation occurred is indicated on the y-axis as yes (Y) or no (N), by SEE isolates. Circles represent acute clinical isolates, and triangles represent inapparent carrier isolates. (B) Sites of methylation (x-axis), by the methylation motif (y-axis). The type of methylation and exact position in the genome are indicated by different colors. Circles represent acute clinical isolates, and triangles represent inapparent carrier isolates.

consistently differed between the carrier and acute clinical isolates of SEE (Figs 3A and 4A). Using REBASE, we performed comparisons of the identified motifs to those in the reference genome, SEE 4047, in which REBASE used with the GenBank data for SEE 4047 to predict restriction enzyme and DNA methyltransferase genes [37]. We identified novel methylation motifs from the complete methylomes of the Swedish SEE isolates. The first new motif (ANNNGANCGNNNAATNNT) was associated with the m6A modification found in a clinical and a carrier SEE isolate, 470_001 and 470_008, respectively (Table 2). The second new motif (DNRTGCAGB) was observed in 4 carrier SEE isolates at 3 locations with the m6A type modification (Fig 3B); although we found other sites with this motif, we were unable to

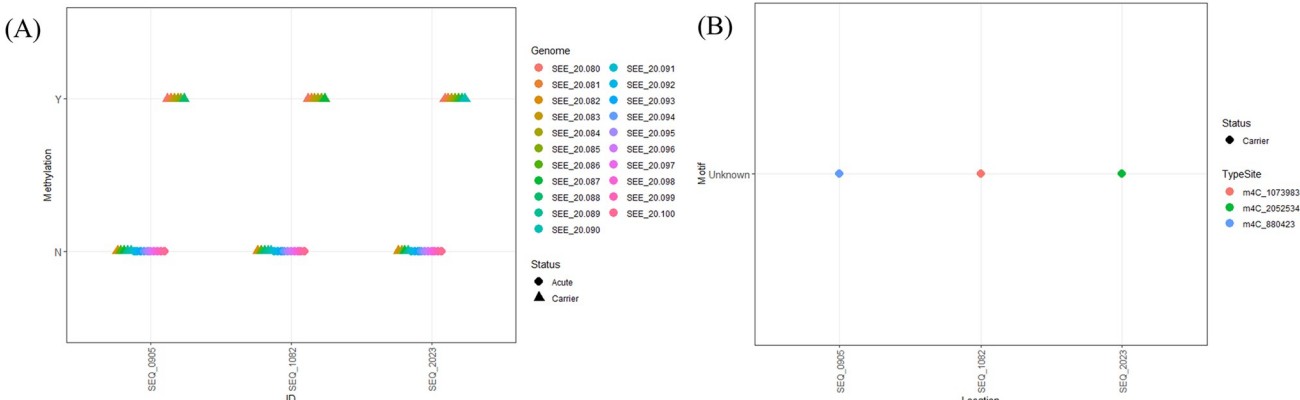

**Fig 4. Methylation locations and motifs from SEE isolates from Pennsylvania.** (A) Depiction of whether methylation occurred at a specified genomic location. Genomic locations are indicated along the x-axis, and whether methylation occurred is indicated on the y-axis as yes (Y) or no (N) by SEE isolates. Circles represent acute clinical isolates, and triangles represent inapparent carrier isolates. (B) Sites of methylation (x-axis), by the methylation motif (y-axis). The type of methylation and exact position in the genome correspond to the different colors. Circles represent acute clinical isolates, and triangles represent inapparent carrier isolates.

**Table 2. The summary of the methylation motif sequences, modification types, and modification percentage for all study SEE isolates from Sweden and Pennsylvania.**

| Genome ID | Location | Status | Motif Sequence | Modification Type | Percent Modification |
|---|---|---|---|---|---|
| 470_001 | Sweden | Acute | ANNNGANCGNNNAATNNT | m6A | 0.86 |
| 470_001 | Sweden | Acute | CATCC | m6A | 0.99 |
| 470_001 | Sweden | Acute | CTGCAG | m6A | 0.97 |
| 470_002 | Sweden | Acute | CATCC | m6A | 0.98 |
| 470_002 | Sweden | Acute | CTGCAG | m6A | 0.97 |
| 470_003 | Sweden | Carrier | CATCC | m6A | 0.98 |
| 470_003 | Sweden | Carrier | CTGCAG | m6A | 0.97 |
| 470_003 | Sweden | Carrier | DNRTGCAGB | Modified Base | 0.42 |
| 470_006 | Sweden | Acute | CATCC | m6A | 0.98 |
| 470_006 | Sweden | Acute | CTGCAG | m6A | 0.97 |
| 470_007 | Sweden | Carrier | CATCC | m6A | 0.99 |
| 470_007 | Sweden | Carrier | CTGCAG | m6A | 0.97 |
| 470_008 | Sweden | Carrier | ANNNGANCGNNNAATNNT | m6A | 0.85 |
| 470_008 | Sweden | Carrier | CATCC | m6A | 0.98 |
| 470_008 | Sweden | Carrier | CTGCAG | m6A | 0.97 |
| 489_001 | Sweden | Acute | CATCC | m6A | 0.99 |
| 489_001 | Sweden | Acute | CTGCAG | m6A | 0.97 |
| 489_002 | Sweden | Acute | CATCC | m6A | 0.99 |
| 489_002 | Sweden | Acute | CTGCAG | m6A | 0.97 |
| 489_003 | Sweden | Acute | CATCC | m6A | 0.99 |
| 489_003 | Sweden | Acute | CTGCAG | m6A | 0.97 |
| 489_004 | Sweden | Acute | CATCC | m6A | 0.99 |
| 489_004 | Sweden | Acute | CTGCAG | m6A | 0.97 |
| 489_005 | Sweden | Carrier | CATCC | m6A | 0.99 |
| 489_005 | Sweden | Carrier | CTGCAG | m6A | 0.97 |
| 489_005 | Sweden | Carrier | DNRTGCAGB | Modified Base | 0.51 |
| 489_006 | Sweden | Carrier | CATCC | m6A | 0.99 |
| 489_006 | Sweden | Carrier | CTGCAG | m6A | 0.97 |
| 489_007 | Sweden | Carrier | CATCC | m6A | 0.99 |
| 489_007 | Sweden | Carrier | CTGCAG | m6A | 0.97 |
| 489_009 | Sweden | Carrier | CATCC | m6A | 0.99 |
| 489_009 | Sweden | Carrier | CTGCAG | m6A | 0.97 |
| 489_009 | Sweden | Carrier | DNRTGCAGB | Modified Base | 0.57 |
| 489_010 | Sweden | Carrier | CATCC | m6A | 0.98 |
| 489_010 | Sweden | Carrier | CTGCAG | m6A | 0.96 |
| 489_010 | Sweden | Carrier | DNRTGCAGB | Modified Base | 0.49 |
| 20–080 | Pennsylvania | Carrier | CTGCAG | m6A | 0.95 |
| 20–081 | Pennsylvania | Carrier | CTGCAG | m6A | 0.95 |
| 20–082 | Pennsylvania | Carrier | CTGCAG | m6A | 0.95 |
| 20–083 | Pennsylvania | Carrier | CTGCAG | m6A | 0.95 |
| 20–084 | Pennsylvania | Carrier | CATCC | m6A | 0.98 |
| 20–084 | Pennsylvania | Carrier | CTGCAG | m6A | 0.97 |
| 20–085 | Pennsylvania | Carrier | CATCC | m6A | 0.98 |
| 20–085 | Pennsylvania | Carrier | CTGCAG | m6A | 0.97 |
| 20–086 | Pennsylvania | Carrier | CTGCAG | m6A | 0.95 |
| 20–087 | Pennsylvania | Carrier | CTGCAG | m6A | 0.95 |

*(Continued)*

**Table 2.** (Continued)

| Genome ID | Location | Status | Motif Sequence | Modification Type | Percent Modification |
|---|---|---|---|---|---|
| 20–088 | Pennsylvania | Carrier | CATCC | m6A | 0.98 |
| 20–088 | Pennsylvania | Carrier | CTGCAG | m6A | 0.97 |
| 20–089 | Pennsylvania | Carrier | CATCC | m6A | 0.98 |
| 20–089 | Pennsylvania | Carrier | CTGCAG | m6A | 0.97 |
| 20–090 | Pennsylvania | Carrier | CTGCAG | m6A | 0.95 |
| 20–091 | Pennsylvania | Acute | CATCC | m6A | 0.97 |
| 20–091 | Pennsylvania | Acute | CTGCAG | m6A | 0.95 |
| 20–092 | Pennsylvania | Acute | CATCC | m6A | 0.97 |
| 20–092 | Pennsylvania | Acute | CTGCAG | m6A | 0.96 |
| 20–093 | Pennsylvania | Acute | CTGCAG | m6A | 0.95 |
| 20–094 | Pennsylvania | Acute | CATCC | m6A | 0.98 |
| 20–094 | Pennsylvania | Acute | CTGCAG | m6A | 0.97 |
| 20–095 | Pennsylvania | Acute | CTGCAG | m6A | 0.95 |
| 20–096 | Pennsylvania | Acute | CATCC | m6A | 0.98 |
| 20–096 | Pennsylvania | Acute | CTGCAG | m6A | 0.97 |
| 20–097 | Pennsylvania | Acute | CATCC | m6A | 0.98 |
| 20–097 | Pennsylvania | Acute | CTGCAG | m6A | 0.97 |
| 20–098 | Pennsylvania | Acute | CATCC | m6A | 0.98 |
| 20–098 | Pennsylvania | Acute | CTGCAG | m6A | 0.97 |
| 20–099 | Pennsylvania | Acute | CATCC | m6A | 0.98 |
| 20–099 | Pennsylvania | Acute | CTGCAG | m6A | 0.97 |
| 20–099 | Pennsylvania | Acute | GGATGH | m6A | 0.21 |
| 20–100 | Pennsylvania | Acute | CATCC | m6A | 0.98 |
| 20–100 | Pennsylvania | Acute | CTGCAG | m6A | 0.97 |

m6A, N6-methyl-adenosine.

determine whether they were either m6A or m4C methylation (*i.e.*, Modified Base; Table 2). The most common motif seen among all SEE isolates regardless of location was CTGCAG (Table 2), which was associated with a type II restriction enzyme and methyltransferase according to REBASE. We also observed that the motif CATCC was found among all Swedish SEE isolates, but only in 12 of the 21 PA-USA SEE isolates (Table 2). Specific methylation sites that occurred in at least half the isolates for either disease status were considered. Six sites were identified that fit this criterion in the SEE isolates from Sweden; 2 of these were identified in acute clinical isolates and the remaining 4 were identified in carrier isolates (Fig 3A, S5 Table). Within the genomes of the PA-USA SEE isolates, only 3 sites of methylation occurred in half of the carrier group (Fig 4A), and these sites all had m4C type modification with an unknown motif (Fig 4B, S6 Table).

To assess differences in gene expression determined by RNA-Seq between acute clinical and inapparent carrier strains of SEE from PA-USA, we used a similar untargeted approach as was used to analyze the SEE isolate exhibiting phenotype switching among colonies [25]. Our differential gene expression analysis with edgeR did not identify any genes that were significantly (FDR ≤ 0.05, logFC ≤ -1 or ≥ 1) differentially expressed (Fig 5). Two genes (SEQ_0823, SEQ_0834) that were closest to being significant and that had a logFC < -1 and an FDR of 0.055 (S2 Fig, S7 Table) were associated with phage elements of the SEE genome found in the prophage φSeq2 in SEE 4047, but these elements have not been further studied.

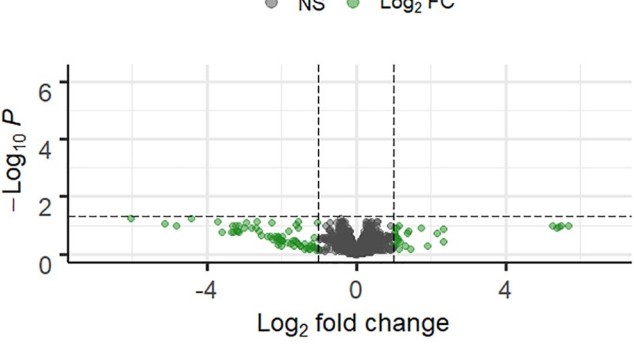

**Fig 5. Volcano plot of Pennsylvania SEE RNA-Seq genes counts.** The $\log_2$ fold-change (logFC) is represented along the x-axis, and the $\log_{10}$-transformed false discovery rate (FDR) is represented along the y-axis. Gray points represent genes that were not identified as significantly differentially expressed (FDR $\leq$ 0.05), and green points represent genes whose expression had a logFC $\leq$ -1 or $\geq$ 1. No genes met the criteria for interest of having an FDR $\leq$ 0.05 and a logFC $\leq$ -1 or $\geq$ 1.

## Discussion

Comparing the AGE, methylomes, and transcriptomes of SEE isolates from horses either with acute clinical signs or that were inapparently-infected and that were derived from outbreaks in 2 different continents, we could not identify significant differences between strains of SEE from acute clinical and inapparent carrier strains. The genomic analysis of the AGE of the PA-USA and Swedish strains were considered separately to avoid confounding effects of geographical origin on any observed genomic differences between acute and carrier strains, because geographical clustering has previously been identified [10, 14]. We defined the core genome for all isolates within a region, which differs from the approach taken in another study where the core genome was delineated by removing prophages, and the ICEs- from the SEE 4047 genome, and any regions of other SEE genomes > 200 base-pairs that did not match the core genome were considered as part of the accessory genome [10]. Harris et al. demonstrated that the prophages φSeq2–4, and ICE*Se1* and ICE*Se2* were highly conserved among SEE isolates [10]. This finding led us to adapt our AGE definition to include all elements found present in some genomes but absent from others, therefore allowing these prophages and ICEs sequences to be considered as elements of the core genome.

The Swedish SEE isolates (n = 14) were collected throughout a single outbreak that occurred among Icelandic horses as previously reported [13]. Of note, 5 of the horses had isolates from both acute disease and after becoming inapparent carriers. Our 2nd population of SEE isolates (n = 21) were collected from a single region of PA-USA, but were not from a single outbreak. We observed fewer AGE for the Swedish SEE isolates than the PA-USA SEE isolates. The Swedish isolates were collected from a single outbreak and the homogeneity of the Swedish isolates indicates that the outbreak was the result of a single SEE strain. This finding is important because it demonstrates the persistence for over a year of a single strain of SEE in a closed herd. Additionally, the Swedish SEE genomes were more continuous (median number of contigs, 1; range, 1 to 4 contigs) than the genomes of the PA-USA isolates (median number of contigs, 4; range, 1 to 11 contigs) (S2 Table). Among both sets of isolates, we observed no

consistent differences in the AGE from isolates collected from inapparent carrier horses when compared to those collected from individuals exhibiting acute clinical signs. These finding are consistent with those previously described, despite the aforementioned difference between studies in how the accessory genome was defined [10]. Many of the observed AGE from these isolates were related to phages, ICEs, and hypothetical proteins that have not been characterized or identified in the reference genome SEE 4047 (S4 Table). These findings are important because they indicate that strains of SEE from inapparent carriers cannot be distinguished by the presence or absence of any specific genes. However, the AGE from SEE isolates collected from different regions of the world do differ [10], regardless of disease status, which illustrates the importance of accounting for geographical effects when comparing genomes of SEE. It should be noted that most 6/8; S1 Table) of the Swedish carrier isolates from this study [13, 43] were collected from individuals that were healthy and lacked evidence of abnormalities including chondroids in their guttural pouches, findings that have been identified in some inapparent carriers [9, 11]. Among the PA-USA carrier isolates, all 5 of the 11 carrier horses for which history was available had gross abnormalities observed via endoscopy in their guttural pouches; endoscopic findings of the remaining 6 horses were not available to the authors (S1 Table). This difference likely reflects differences between following horses from a single herd over a period of time (Swedish isolates) and identifying individual horses from multiple locations that were shedding SEE ≥ 6 weeks after recovering from clinical signs. Although other studies have described potential pathogen-associated genetic changes that could result in a carrier state of SEE [10, 15, 16], we did not identify truncation of the SeM protein in any of the carrier strains from PA-USA (n = 11) and only truncation in 2 of 8 Swedish carrier strains (489_006, 489_010) as previously described [13], and the equibactin locus was found in all SEE strains from both locations (*i.e.*, acute clinical and carrier strains). Although the reasons for the discrepancy between our findings and prior studies is unknown, it is possibly attributable to either geographical differences or clinical phenotypic differences (*e.g.*, isolates obtained from horses with chondroids or guttural pouch empyema [15]) between isolates in our study and isolates from previous studies. The variations within the Swedish SEE isolates collected from the same horse, such as the truncated SeM protein observed in 2 carrier isolates (S1 Fig), likely reflect adaptation of SEE to its host over time. These variations were noted between the 2 disease states (acute clinical or inapparent carrier) and source (guttural pouch or nasopharyngeal lavage; Table 1; S1 Fig). Nevertheless, these pathogen-adaptions cannot fully explain the carrier state because they were not consistent among the SEE isolates from Sweden from within the same horse or from other inapparent carrier isolates of SEE from Sweden or PA-USA.

PacBio WGS results also were used to describe the methylome of the PA-USA and Swedish SEE isolates. Methylation in prokaryotes has primarily been described as a mechanism of defense against invading bacteriophages and other foreign DNA [20, 21]. Lack of methylation at a particular motif that occurs throughout the genome has been shown to produce modifications in the gene expression of microbes [22, 24], even contributing to the virulence of some pathogens [22]. To the authors' knowledge, this is the first detailed comparison of the global methylomes of inapparent carrier and acute clinical isolates of SEE. Despite the more comprehensive scrutiny of genetic elements of our approach, we failed to identify any changes in methylation that differentiated between inapparent carrier and acute clinical isolates of SEE (Figs 3 and 4). As we observed for the AGE analyses, methylation patterns differed between the geographical regions. Among the methylation observed in the SEE strains from Sweden, we identified 2 novel motifs that have not been described previously in SEE isolates. The lower frequency of methylation events associated with the novel motif (DNRTGCAGB) increases the likelihood that the absence of methylation at this motif could influence the gene expression in

these isolates. Decreased methylation of target motifs has been reported to inhibit *Streptococcus pyogenes* from surviving in human neutrophils and to reduce expression of genes involved in immune evasion and adherence [22], to alter the ability of *Borrelia burgdorferi* to colonize the host [24], and to alter the expression of genes associated with metabolic pathways of *Mycobacterium tuberculosis* [23]. To evaluate sites with higher prevalence of methylation, we considered sites in which methylation occurred in at least half the isolates from either the carrier group or the acute clinical disease group. Six methylation sites were identified as occurring in at least half of the Swedish isolates of SEE, and only 3 methylation sites were identified in at least half of the PA-USA isolates of SEE (Figs 3 and 4). Of the 9 sites that had more frequent methylation in both geographical locations of SEE isolates, none were common among isolates from both locations. This further demonstrates geographical differences in the methylome of SEE, but methylation patterns did not differ between the 2 different phenotypes. Little can be inferred about the biological effects of the observed differences in methylation between geographical areas without further investigations, but our objective was to determine whether methylation patterns differed consistently between isolates of SEE from inapparent carriers and acute clinical disease.

RNA-Seq has been previously used to assess gene expression in SEE isolates [10, 25]. Changes in transcription identified using untargeted RNA-Seq of an SEE isolate were associated with a difference in the phenotype of colonies of the isolate [25]. A targeted approach to gene expression (*viz.*, quantitative PCR) was used to evaluate gene expression of the *has* operon which regulated levels of hyaluronic acid capsule expression in SEE isolates where deletions in the *has* operon were identified [10]. We performed untargeted RNA-Seq on inapparent carrier and acute clinical SEE isolates from the same region of PA-USA. No significantly (FDR $\leq$ 0.05) differentially expressed genes were identified between the acute and carrier SEE isolates (S7 Table). We did identify, however, 2 CDS, SEQ_0823 and SEQ_0843, that were closest to fitting our defined criteria for significance and magnitude of effect, and both were associated with mobile genetic elements found in the prophage φSeq2 of the SEE 4047 genome. For both genes, the magnitude of expression was only highly elevated in 3/10 of the acute SEE isolates from PA-USA (S2 Fig). Besides being identified as a putative phage portal protein (SEQ_0823) and putative phage tail protein (SEQ_0843), not much is known about either of these genes. Homologs proteins of SEQ_0823 were found in *Streptococcus pyogenes*, *Streptococcus dysagalactiae*, and *Streptococcus agalactiae* with a similarity of 96%, and for SEQ_0843 in *Streptococcus equi* subsp. *zooepidemicus* with a similarity of 100%.

This study has a number of limitations. First, the definition of inapparent carriers can be highly variable [9, 12–14]. The inapparent carrier strains of SEE from Sweden were recovered from horses between 12 and 50 weeks after resolution of their clinical signs, whereas the PA-USA inapparent carrier horses were collected between 6 and 20 weeks after resolution of clinical signs, or had no clinical signs observed (Table 1). Nevertheless, we found no evidence from horses from either location of any consistent differences in the genome or methylome of these isolates indicating any specific adaptations to the host environment, even among the Swedish strains representing isolates of acute disease and inapparent carrier phenotypes in the same animal. The PA-USA isolates were not all from the same outbreak or the same year, but even among isolates from within farm and year, there were no consistencies observed. Moreover, none of the PA-USA samples were derived from the same animal, and the number of isolates studied was modest. Nonetheless, this is the first comprehensive analysis of the genomes, methylomes, and transcriptomes of inapparent carrier and acute clinical strains of SEE. We only had isolates of PA-USA available to evaluate using RNA-Seq. Another limitation of the RNA-Seq approach was that SEE were grown in liquid media and this might not reflect transcription within the host [44]. However, an effective approach for studying transcription of

SEE within its host's cells remains limited; to the authors' knowledge, this study provides comparisons of untargeted gene expression of carrier and acute SEE isolates from the USA that has not been previously available.

The most important finding from this study is that we failed to identify any consistent or specific pathogen-associated changes between the inapparent carrier strains and the acute clinical disease strains of SEE using a few NGS techniques. Although genomic differences were observed between the 2 geographical regions, no changes in the genome, methylome, or transcriptome were identified that could be interpreted as reflecting a consistent mechanism of adaptation of SEE to the host resulting in inapparent carriage. These findings indicate that host-associated differences are a more likely explanation of the bacterium's ability to persist in horses without resulting in either clinical signs or a robust immune response (*i.e.*, the presentation of clinical disease) [45]. Thus, further evaluation of host immune responses to SEE is warranted to elucidate how to identify and eliminate chronic carriers of SEE to control and prevent this important equine infectious disease.

## Supporting information

**S1 Fig. Phylogenetic tree of 14 SEE isolates from Sweden by horse.** SEE isolates from the outbreak did not cluster by the individual horse from which the isolate was collected, but results demonstrate variation of isolates recovered from the same individual over time. [a]Denotes truncation in the SeM protein; GPL, Guttural pouch lavage; NL, Nasopharyngeal lavage; SeM, M-like protein.
(TIF)

**S2 Fig. RNA-Seq expression values for 2 SEE genes by disease status group.** (A) Expression level (y-axis) of SEQ_0823 by disease presentation (x-axis). Only 3/10 of the acute SEE isolates had elevated expression levels. (B) Expression level (y-axis) of SEQ_0834 by disease presentation (x-axis). Only 3/10 of the acute SEE isolates had higher expression levels.
(TIF)

**S1 Table. Guttural pouch endoscopy findings of inapparent carrier horses from Sweden (n = 8) and Pennsylvania, USA (n = 11).**
(XLSX)

**S2 Table. Genome accession numbers for SEE isolates from Sweden and Pennsylvania.**
(XLSX)

**S3 Table. Annotation and bin location for the accessory genome elements for the SEE isolates from Sweden.**
(XLSX)

**S4 Table. Annotation and bin location for the accessory genome elements for the SEE isolates from Pennsylvania.**
(XLSX)

**S5 Table. Sites of methylation found in at least half (n $\geq$ 4) of either disease state in Swedish SEE isolates.**
(XLSX)

**S6 Table. Sites of methylation found in at least half (n $\geq$ 6) of either disease state in Pennsylvania SEE isolates.**
(XLSX)

**S7 Table. Transcriptome-wide gene expression identified by *edgeR* analysis.**
(XLSX)

**S1 Appendix. Linux and R code used for accessory genome, methylome, and transcriptome analysis.**
(TXT)

**S2 Appendix. Clustal OMEGA multiple sequence alignment of the SeM protien from SEE isolates from Sweden (n = 14) and Pennsylania (n = 21).**
(TXT)

## Acknowledgments

We would like to thank Dr. Bibiana Petri da Silveira, Texas A&M University for her technical assistance, and Ms. Sheri Young from the University of Pennsylvania for her assistance in culturing the isolates from Pennsylvania. For the Swedish outbreak, Karin Nygren, Kamilla Ekström, and Cecilia Ekberg are gratefully thanked for their assistance during sample collection throughout the study. Portions of this research were conducted with the advanced computing resources provided by Texas A&M High Performance Research Computing.

## Author Contributions

**Conceptualization:** Ellen Ruth A. Morris, Ashley G. Boyle, Miia Riihimäki, Angela I. Bordin, John Pringle, Noah D. Cohen.

**Data curation:** Ellen Ruth A. Morris, Miia Riihimäki, Anna Aspán, Andrew E. Hillhouse, John Pringle, Noah D. Cohen.

**Formal analysis:** Ellen Ruth A. Morris, Miia Riihimäki, Andrew E. Hillhouse, Ivan Ivanov, Noah D. Cohen.

**Funding acquisition:** John Pringle, Noah D. Cohen.

**Investigation:** Ellen Ruth A. Morris, Ashley G. Boyle, Miia Riihimäki, Anna Aspán, Eman Anis, Andrew E. Hillhouse, John Pringle, Noah D. Cohen.

**Methodology:** Ellen Ruth A. Morris, Ashley G. Boyle, Miia Riihimäki, Anna Aspán, Andrew E. Hillhouse, Noah D. Cohen.

**Project administration:** Noah D. Cohen.

**Resources:** Ashley G. Boyle, Miia Riihimäki, Anna Aspán, Eman Anis, John Pringle, Noah D. Cohen.

**Supervision:** Andrew E. Hillhouse, Ivan Ivanov, Angela I. Bordin, John Pringle, Noah D. Cohen.

**Validation:** Ashley G. Boyle, Miia Riihimäki, Andrew E. Hillhouse.

**Visualization:** Ellen Ruth A. Morris.

**Writing – original draft:** Ellen Ruth A. Morris, Noah D. Cohen.

**Writing – review & editing:** Ellen Ruth A. Morris, Ashley G. Boyle, Miia Riihimäki, Anna Aspán, Eman Anis, Andrew E. Hillhouse, Ivan Ivanov, Angela I. Bordin, John Pringle, Noah D. Cohen.

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
