## [Decision Letter · Decision Letter 0]

7 May 2021

PONE-D-21-07625

Differences in the Genome, Methylome, and Transcriptome Do Not Differentiate Isolates of Streptococcus equi subsp. equi from Horses with Acute Clinical Signs from Isolates of Inapparent Carriers

PLOS ONE

Dear Dr. Cohen,

Thank you for submitting your manuscript to PLOS ONE. After careful consideration, we feel that it has merit but does not fully meet PLOS ONE’s publication criteria as it currently stands. Therefore, we invite you to submit a revised version of the manuscript that addresses the points raised during the review process.

Your manuscript has been reviewed by an expert in your field. The reviewer did have two questions:

Line 116: were any or all of the carriers examined by endoscopy to determine whether guttural pouch pathology was present? If so, it would be helpful to provide further information and to identify which carriers had endoscopically visible guttural pouch pathology. This point is picked up in the discussion (carrier isolates being obtained from animals with and without gross pouch abnormalities) and it would be useful to have more details provided.

Line 258: This is an interesting observation – could the authors speculate on possible reasons for this in the discussion? For example, whether this is potentially due to the Swedish isolates all being obtained from a single outbreak and over a shorter time frame?

Please answer these questions and make the necessary revision.  

We look forward to receiving your revised manuscript.

Kind regards,

Yung-Fu Chang

Academic Editor

PLOS ONE

Journal Requirements:

We note that you have stated that you will provide repository information for your data at acceptance. Should your manuscript be accepted for publication, we will hold it until you provide the relevant accession numbers or DOIs necessary to access your data. If you wish to make changes to your Data Availability statement, please describe these changes in your cover letter and we will update your Data Availability statement to reflect the information you provide.

Reviewers' comments:

Reviewer's Responses to Questions

**Comments to the Author**

1. Is the manuscript technically sound, and do the data support the conclusions?

Reviewer #1: Yes

2. Has the statistical analysis been performed appropriately and rigorously? 

Reviewer #1: N/A

3. Have the authors made all data underlying the findings in their manuscript fully available?

Reviewer #1: Yes

4. Is the manuscript presented in an intelligible fashion and written in standard English?

Reviewer #1: Yes

5. Review Comments to the Author

Reviewer #1: This is a well written, well designed and interesting paper that addresses an important question about SEE that has been debated for some years: to what extent is the clinical phenotype switch from acute disease to chronic, asymptomatic due to bacterial factors. Previous studies have found some genetic differences between strains isolated from acute cases and those from carriers, leading to the suggestion that pathogen-associated genetic changes represent a form of immune evasion, or result in other pathogen phenotypic changes, that enable the carrier state. This study design is intelligent and compares isolates from acute and carrier cases from an outbreak in Sweden with isolates from acute and carrier cases from outbreaks in the USA (Pennsylvania) for differences at DNA, methylated DNA and RNA levels. The matching of acute and carrier isolates from the same region addresses the geographical variation in SeM and ST that has been previously reported.

The abstract accurately reflects the content of the paper and its conclusions.

The introduction is complete and provides a clear justification and rationale for the study.

The methods are clearly described. The results are well and clearly presented and the tables and figures provided are useful. The discussion is balanced and addresses the points of interest in the results, including

The section discussing limitations is well constructed and addresses the intrinsic difficulties and limitations in defining carriers, the difference in duration of carriage between the Swedish and USA isolates, the limitations of the USA collection of isolates. The authors also make some useful comments about what can be inferred from these limitations, which is helpful to readers. The authors also acknowledge the inherent challenges of performing transcriptomics on isolates grown in vitro due to the alterations in gene expression compared with the in vivo state.

The conclusions are supported by the data presented in the paper.

Overall, this is useful and important study which adds significant new information to the field by addressing a question that is central to SEE pathogenesis.

Specific questions:

Line 116: were any or all of the carriers examined by endoscopy to determine whether guttural pouch pathology was present? If so, it would be helpful to provide further information and to identify which carriers had endoscopically visible guttural pouch pathology. This point is picked up in the discussion (carrier isolates being obtained from animals with and without gross pouch abnormalities) and it would be useful to have more details provided.

Line 258: This is an interesting observation – could the authors speculate on possible reasons for this in the discussion? For example, whether this is potentially due to the Swedish isolates all being obtained from a single outbreak and over a shorter time frame?

6. PLOS authors have the option to publish the peer review history of their article (what does this mean?). If published, this will include your full peer review and any attached files.

Reviewer #1: No

---

## [Author Response · Author response to Decision Letter 0]

19 May 2021

May 19, 2021

Prof. Yung-Fu Chang

Academic Editor

PLoS One

Subject: Revision of manuscript entitled, “Differences in the Genome, Methylome, and Transcriptome Do Not Distinguish Isolates of Streptococcus equi subsp. equi from Horses with Acute Clinical Signs from Isolates of Inapparent Carriers” 

Dear Dr. Chang:

Thank you for your electronic message dated May 7, 2021 about the above-referenced manuscript. I also thank the reviewers who carefully considered our report. My coauthors and I have revised the report on the basis of the reviewers’ comments. Below, we detail how each point raised by the reviewers was addressed in the revised report. The revised report with marked changes and an unmarked manuscript will be uploaded to the PLoS One website along with this response letter. 

Comments from Reviewer 1:

Line 116: were any or all of the carriers examined by endoscopy to determine whether guttural pouch pathology was present? If so, it would be helpful to provide further information and to identify which carriers had endoscopically visible guttural pouch pathology. This point is picked up in the discussion (carrier isolates being obtained from animals with and without gross pouch abnormalities) and it would be useful to have more details provided.

AUTHORS’ RESPONSE: We thank the reviewer for raising this important point. All of the carrier horses were examined endoscopically. Unfortunately, follow-up information with details was not determined for 6 of the 11 horses from Pennsylvania. We have revised the report to include a table summarizing these findings (new Supplemental Table 1), and also included some discussion of these findings. 

Line 258: This is an interesting observation – could the authors speculate on possible reasons for this in the discussion? For example, whether this is potentially due to the Swedish isolates all being obtained from a single outbreak and over a shorter time frame?

AUTHORS’ RESPONSE: We thank the reviewer for raising another great point. We think that the Swedish isolates had more similar (fewer) AGE because they were from a single outbreak. Interestingly, they were actually collected over a period of > 1 year, which demonstrates persistence of a single strain in horses without clinical signs for over 1 year (albeit with some variation including truncation of SeM). These isolates also had fewer contigs which might have contributed to more consistent results for isolates from the same outbreak. We have revised the report to address these points.

Again, we appreciate your efforts and those of the reviewers. Please let us know if you have any questions or concerns regarding our revised report.

---

## [Editor Report · Decision Letter 1]

24 May 2021

Differences in the Genome, Methylome, and Transcriptome Do Not Differentiate Isolates of Streptococcus equi subsp. equi from Horses with Acute Clinical Signs from Isolates of Inapparent Carriers

PONE-D-21-07625R1

Dear Dr. Cohen,

We’re pleased to inform you that your manuscript has been judged scientifically suitable for publication and will be formally accepted for publication once it meets all outstanding technical requirements.

Kind regards,

Yung-Fu Chang

Academic Editor

PLOS ONE
---

## [Editor Report · Acceptance letter]

4 Jun 2021

PONE-D-21-07625R1 

Differences in the Genome, Methylome, and Transcriptome Do Not Differentiate Isolates of *Streptococcus equi* subsp. *equi* from Horses with Acute Clinical Signs from Isolates of Inapparent Carriers 

Dear Dr. Cohen:

I'm pleased to inform you that your manuscript has been deemed suitable for publication in PLOS ONE. Congratulations! Your manuscript is now with our production department. 

Kind regards, 

on behalf of

Dr. Yung-Fu Chang 

Academic Editor

PLOS ONE